# Nanoindentation of Multifunctional Smart Composites

**DOI:** 10.3390/polym14142945

**Published:** 2022-07-20

**Authors:** Zhenxue Zhang, Denise Bellisario, Fabrizio Quadrini, Simon Jestin, Francesca Ravanelli, Mauro Castello, Xiaoying Li, Hanshan Dong

**Affiliations:** 1School of Metallurgy and Materials, University of Birmingham, Birmingham B15 2TT, UK; z.zhang.1@bham.ac.uk (Z.Z.); x.li.1@bham.ac.uk (X.L.); h.dong.20@bham.ac.uk (H.D.); 2Department of Industrial Engineering, University of Rome ‘Tor Vergata’, 00133 Rome, Italy; fabrizio.quadrini@uniroma2.it; 3CANOE, Le Centre Technologique Nouvelle Aquitaine Composites & Matériaux Avancés, Bât CHEMINNOV—ENSCBP, 33600 Pessac, France; jestin@plateforme-canoe.com; 4Elica S.p.A., Via Ermanno Casoli, 2, 60044 Fabriano, Italy; f.ravanelli@elica.com (F.R.); m.castello@elica.com (M.C.)

**Keywords:** nanoindentation, nanocomposite, interfacial shear strength, shape memory polymer, carbon fibre-reinforced composite

## Abstract

Three multifunctional smart composites for next-generation applications have been studied differently through versatile nanoindentation investigation techniques. They are used in order to determine peculiarities and specific properties for the different composites and to study the charge/matrix, charge/surface, or smart functions interactions. At first, a mapping indentation test was used to check the distribution of hardness and modulus across a large region to examine any non-uniformity due to structural anomalies or changes in properties for a carbon nanotubes (CNTs)-reinforced polypropylene (PP V-2) nanocomposite. This smart composite is suitable to be used in axial impeller fans and the results can be used to improve the process of the composite produced by injection moulding. Secondly, the interfacial properties of the carbon fibre (CF) and the resin were evaluated by a push-out method utilizing the smaller indentation tip to target the individual CF and apply load to measure its displacement under loads. This is useful to evaluate the effectiveness of the surface modification on the CFs, such as sizing. Finally, nanoindentation at different temperatures was used for the probing of the in situ response of smart shape memory polymer composite (SMPC) usable in grabbing devices for aerospace applications. Furthermore, the triggering temperature of the shape memory polymer response can be determined by observing the change of indentations after the heating and cooling cycles.

## 1. Introduction

Carbon nanotubes (CNTs) are light, strong, flexible, and thermally and electrically conductive. An interesting route toward multifunctional composite materials is through the introduction of CNTs into carbon fibre-reinforced polymers (CFRP) by modifying either the polymer matrix or the carbon fibres (CFs) as this may not only provide integrated sensing capability but also lead to some additional mechanical reinforcement [1]. Therefore, integrating CNTs into polymer attracts wide interest in developing smart composites such as sensors, conductive polymers, and energy storage and conversion devices etc. [2]. CFs have high stiffness, high tensile strength, high chemical resistance, and low thermal expansion. In a CFRP, CFs provide the strength and stiffness to reinforce polymer, while polymer bestows a cohesive matrix to preserve the CFs together and enhance their toughness. However, the bonding between CFs and the polymer is still an issue. Approaches, such as plasma treatment and sizing CFs with carbon nanofillers, have been used to improve the adhesion between CFs and the resin [3,4]. Shape memory polymers (SMP) have large recoverable strains, low cost, good manufacturability, and they are being actively considered for applications in many fields such as in automotive and aerospace structures, biomedical devices, and microsystems [5]. However, pure SMPs are brittle in their glassy and rubbery state, which compromise their reliability [6]. The emerging development of CFs-reinforced shape memory polymer composites (SMPCs) has improved the inherent poor mechanical properties of SMPs while retaining their large recoverable deformation [7]. 

A nanoindentation test uses electromagnetic force and capacitive depth measurement to evaluate the elastic and plastic properties of materials on the nanoscale. It is a versatile method to characterize a wide range of materials, especially for a homogeneous continuum system. Over the last few decades, polymers filled with functional reinforcements such as CFs and nanoparticles (i.e., CNTs, CNFs, graphene) have received considerable attention and more widespread application. The extremely small force and displacement resolutions allow the nanoindentation to characterize the inhomogeneous composite materials composed of discrete regions with distinct material properties. The application of nanoindentation in the characterization of polymer composite materials has been a fast-growing research area as briefed in Table 1. For example, by conducting multiple nanoindentations in a defined grid pattern (often referred to as grid indentation or modulus mapping [8]) on a composite specimen, it is possible to determine the elastic modulus and the hardness of either a homogeneous composite or the corresponding properties of its heterogeneous constituents [9]. This can not only compare the nanomechanical properties of different composites to optimize the manufacturing procedures [10] but also to quantitatively determine the interphase size and stiffness of nanofiller on the interfacial properties in CF/epoxy composites [11]. 

Pyramidal indenters such as Berkovich and Vickers are often used for the characterization of metallic and composite materials, and conical and cylindrical punches with flat-ended cylindrical geometry are also frequently used for characterizing the polymers [9,20,21]. For example, a flat-top cylinder indenter was adopted to study the effect of moisture absorption on the reduction of the stiffness of autoclave-cured CFRP plates due to the different ageing in water [13]. The interfacial properties of fibre and resin often significantly influence the performance of the carbon fibre-reinforced composites, and therefore different approaches such as single fibre directly loaded and matrix externally loaded methods have been used to evaluate the interfacial interaction between fibres and matrices [22,23]. Among them, the push-out test (POT) is favoured because it is carried out directly on individual CFs in the composite specimen, and the quantitative values of the interfacial shear strength (IFSS) can be calculated [17,24]. The dynamics of nanoscale strain storage in a shape memory polymer composite (SMPC) was studied using a heated atomic force microscope cantilever technique [25], and the indentation of the shape memory layer was used for assessing their deformation [19]; however, there is little reporting on using nanoindentation to evaluate its in situ response, especially at elevated temperatures.

In this paper, we highlighted the ways we characterized smart composites with different components in one nanoindentation instrument. Firstly, we used the commonly adopted mapping nanoindentation test to look at the distribution of hardness and reduced modulus across a large region of a CNTs-reinforced PPV-2 nanocomposite (CNTs-PPV2) to examine any non-uniformity due to structural anomalies or changes in properties to improve the injection moulding quality. Secondly, we employed a cone-shaped indenter to assess the interfacial properties between the CFs and the resin of the CFRP via a push-out (POT) method aiming to improve their bond via sizing the CFs with different agents with or without carbon nanofillers. In the meantime, we have also initiated a strategy to examine the CFs in different locations to validate the technique. Finally, nanoindentation was manipulated for the probing of the in situ response of a smart shape memory polymer composite (SMPC) at different temperatures. We also developed a technique to identify the triggering temperature of the shape memory response by examining the change of indentations after the heating and cooling cycles. 

## 2. Materials and Methods

### 2.1. CNTs-Reinforced PPV-2 Nanocomposite (CNTs-PPV2)

A CNTs-reinforced PPV-2 nanocomposite (CNTs-PPV2) based on PP V-2 homopolymer (POLAD A 121 BLACK) reinforced with 5 wt% CNTs (Arkema C100 Graphistrength, with mean outer diameter of 10–15 nm and length between 0.1 and 10 µm) was developed to increase the sensing and responding capability of an innovative conveyor/impeller fan for air treatment applications. A first formulation screening was performed using a micro-compounder (MiniLab II Haake) on a few grams of compounds with various CNT contents as processing parameters. It enabled us to characterize the dispersion state, the electrical conductivity, MFI, and piezo-resistive behaviour (sensing). The chosen nanocomposite formulation was prepared by double extrusion using first a BUSS Kneader for masterbatch preparation followed by dilution onto a twin-screw extruder (LabTECH). Then, 70 × 70 mm standard plates with different thicknesses (0.5, 1.0, 1.5, 2.8 mm) (Figure 1a) were produced using injection moulding intending to simulate an axial impeller Fan (Figure 1b). Three different sections (Figure 1c)—inlet, centre, and edge—were cut from the composite plate with different thicknesses and mounted in Bakelite, and progressively ground and polished for nanomechanical measurements at different locations and thickness of the composite, which were used to evaluate the homogeneity of the materials. The load was 20 mN and the holding time was 5 s. A JEOL 7000 FE SEM was used to characterize the CNTs-reinforced PPV-2 nanocomposite.

### 2.2. Nanomechanical Properties and Interfacial Properties of the CFRP 

T700 SC 31 E CFs (T700) with a diameter of approximately 7 μm from Toray were sized with a formulation containing emulsifiers, anti-static agents, and lubricants as well as a polymeric coupling agent that promotes fibre–matrix interactions and thus creates strong interfaces between the fibre and the matrix resin in the CFRP composite. The sizing solution was obtained by mixing the commercial polyurethane dispersion (MICHELMAN HYDROSIZE U6-01) with/without CNT or CNF fillers dispersed by an anionic surfactant, sodium dodecyl benzenesulfonate (SDBS). Sizing content obtained was up to 2.2 wt%. Sized CFs were mounted into a composite and polished to a mirror-like surface which was used for measuring the nanomechanical properties of the CFs, the resin, and their interface. Afterwards, the polished composite was cut into thin discs with a thickness less than 1 mm (Figure 2a). Both sides of the disc were ground and polished progressively to between 30 and 60 µm thick. The prepared test piece was stuck to a special holder with grooves with a width of 30 µm and a depth of 12 µm and then glued to the standard cylinder support for the push-out test (Figure 2b). Figure 2c demonstrates the distribution of the CFs in the CFRP.

### 2.3. Shape Memory Polymer Composite (SMPC) Sandwich Structure 

The SMPC sandwich structure consisted of two CFRP prepreg layers with the interposition of one SMP foam interlayer. Commercial thermosetting CFRP prepregs (HexPly M49/42%/200PW/CCF-3K) suitable for aerospace applications supplied by Hexcel and shape memory epoxy foam produced with a commercial epoxy SM resin (3M) were used. In particular, the SM epoxy foam was fabricated with a solid-state foaming process reported in a previous work of the authors [26]. The epoxy foam presented a density of 0.41 g/cm^3^, and after being cut, it was inserted as a central layer of 1.2 ± 0.02 mm of thickness between two plies of CFRP to form the SMP composite sandwich structure (Figure 3). The structure thus formed was subsequently consolidated through a compression moulding process inside an aluminium mould over a hot plate at 200 °C for 60 min under an applied pressure of 66.7 kPa. Through this manufacturing process, composite sandwich samples of size 50 × 10 mm^2^ with shape memory interlayers were produced. A Nikon XT H225 with 3 µm focal spot size was used to carry out the 3-D characterization of the SMPC to examine the internal structure and to detect the faults and defects. 

### 2.4. Nanoindentation and Push-Out Test

A NanoTest system (Micro Materials Ltd., Wrexham, UK) was used for the nanoindentation measurements. The system has a very high thermal stability enabling nanoindentation measurements to be performed without thermal drift. The nanoindentation tests were conducted in an environmental enclosure controlled at 22.0 ± 0.2 °C. In a nanoindentation test, a suitable region was first selected for the specific test with the support of microscopy. All the samples were loaded from an initial load of 10 μN to a peak load up to 500 mN at a specified loading rate (i.e., 1 mN/s) and held for a set time (i.e., 5 s). A typical loading and unloading curve can be plotted during the measurement (Figure 4a). The hardness is given by the peak load (*P*_max_) divided by the contact area (A) which was calculated according to the shape of the indenter (i.e., a Berkovich indenter was used in this work) and the final indenter displacement (*h_f_*) after complete unloading [9,27].
H = P_max_/A(1)

The reduced modulus (*E_r_*) can be calculated based on the elastic modulus (*E_i_*), Poisson’s ratio (*v_i_*) and the geometry of the indenter, the Poisson’s ratio of the specimen (*v*), together with the initial unloading contact stiffness [9,27].
(2)$$\frac{1}{E_{r}}=\frac{1-v^{2}}{E}+\frac{1-v_{i}^{2}}{E_{i}}$$

For nanoindentation tests performed at temperature, the configuration provided a heating block attached to the sample holder to heat the sample as shown in the indentation setup (Figure 4b). The samples were heated up at a rate of 1.6 °C/min to the target temperature, i.e., 110 °C. The extremely small force and displacement resolutions, plus a large range of applied forces (0–500 mN) and displacements (0–50 µm) allow the nanoindentation to characterize a wide range of smart composite systems.

A cone-shaped indenter with a tip of 5 µm in diameter was used for the push-out test. The cone shape has a better loading and even contact with the individual fibres. At the beginning of the test, the interested region was selected, and the thickness of the specimen was measured by recording the distance change of the focus from the flat area of the holder to the specimen surface. Secondly, the indenter was gradually applied to the selected CFs on top of the grooves to a maximum load (i.e., 60 mN), which was greater than the critical load and decided experimentally (Figure 4c). As the load increased to a certain level, displacement continuously increased with the load unchanged, which corresponds to the debonding of the fibre, and this is the critical load (P_c_). The average interfacial shear strength (IFSS) at the fibre/matrix interface can be obtained by the load divided by the lateral surface area of the short CF, which can be calculated using the diameter of the CF (d) and the sheet thickness (e) as reported in our earlier work [3].
IFSS = 𝑃_𝑐_/(*π*de)(3)

## 3. Results and Discussion

### 3.1. Nanomechanical Properties of the CNTs-PPV2 Nanocomposite

The CNTs-PPV2 nanocomposite has a gauge factor > 2.5, an electrical conductivity > 30 S/m, and a decrease in the melt flow index value (from 31 mm/10 min in the unfilled POLAD to 8.5 mm/10 min with the CNT inclusions). Some properties of the composite can be found in Table 2. The distribution of CNTs in the CNTs-PPV2 nanocomposite were observed by the fractography SEM on a slice of sample cut from the plate. It was found that the CNTs were evenly dispersed in the composites as shown in Figure 5.

Figure 6 shows the nanohardness and the calculated reduced elastic modulus (Er) at the inlet section for three plates with different thicknesses and for which the minimum distance between the indentations was about 50 μm. For the inlet section, the nanohardness in the composite was quite similar for three plates with different thicknesses, and the thick moulded plate (2.8 mm) had a more scattered distribution while its elastic modulus was slightly lower than that of the other two plates (0.5 mm and 1.8 mm). For the centre section, the nanohardness profiles for the three plates were levelled; however, the elastic modulus tended to reduce with the increase of the thickness (Figure 7). For the edge area, the value of both nanohardness and elastic modulus tended to increase with the thickness of the plates (Figure 8), which could be linked to the increased pressure at the corner during the inject moulding process. The nanomechanical test suggests the impellers produced using the compound injected with a standard machine can obtain a uniform mechanical property by optimizing the process parameters.

### 3.2. Evaluation of the Interfacial Properties of the CFRP Composite

The nanomechanical properties of the CFs, the matrix, and their interface were measured first as shown in Figure 9. CFs had much higher nanohardness (5.52/5.60 GPa) in comparison to the resin (0.38). The hardness measured at the interface of CF and resin was between them. The reduced elastic modulus (Er) had a similar trend (Figure 9b).

The interfacial shear strength (IFSS) between CF and the resin in a CFRP composite was evaluated by the POT. In the POT, the load was applied on the selected single CF using the cone-shaped indenter and a counter plate was used to support the thin disc sample. As shown in Figure 10a, a region of interest was chosen, and individual CFs were identified. Then, a load was gradually applied to each location to a peak load (60 mN) greater than the critical load, which was decided experimentally. The different displacement indicates the different penetration achieved during the test up to the maximum established load. As demonstrated by the loading–unloading curve for the second tested carbon fibre (CF2) (in Figure 11), after an initial non-linear stage for a conformal contact of the indenter and the specimen, the displacement increased linearly with the increment of the load. As the load increased to a certain level (~40 mN), the gradient reduced indicating the CF started sliding, and a larger displacement was observed at a higher load (42.7 mN) corresponding to the debonding of the fibre, which was deemed as the critical load for CF2. The thickness of the specimen was 32 µm and the IFSS for CF2 was 60.7 MPa, and the impression around CF2 after the POT can be seen in Figure 10b. The load vs. displacement curves of the six CFs indicated that they have different bond strengths with the resin. CF2 had fewer neighbouring CFs which might make it easy to slide under an increased load. CF4 started to slide out at a load of 46.5 mN but a large movement was observed at a higher load of 58.9 mN, even though CF4 had less constraint around it. It might be due to the CF4 making contact with the groove edge after it was pushed out, as CF4 was located at a rigid region on the top of the groove edge as indicated in Figure 10a and Figure 12. The IFSS of 66.1 MPa was calculated using a critical load of 46.5 mN rather than the higher load of 58.9 mN. CF6 overcame the resistance of the resin slightly at a load of 38 mN, but the critical load was about 44.2 mN, corresponding to an IFSS of 62.8 MPa due to less restriction at the nearby region. As displayed in Figure 11, a larger displacement was observed for CF6 as it was located in a region on top of the middle of the groove (Figure 12). The specimen disc spanned on the top of the groove, and the middle part was easy to bend when subjected to load, however, this did not affect the critical load value. CF3 and CF5 had more CFs surrounding them, and they were only partially pushed out due to the resistance of the neighbouring fibres and the resin. CF1 was surrounded by five CFs which hindered its movement during the POT, and thus it was not pushed out until the peak load of 60 mN, indicating a higher IFSS. There was hardly any change of CF1 before and after the POT, as shown in Figure 10a,b. Therefore, only CF2, CF4, and CF6 were used to calculate the average IFSS of this region, which was 63.2 ± 2.9 MPa.

Using this technique, the interfacial shear strength of CFs and the resin can be calculated, and the impact of neighbouring CFs and the location of the CFs on their bond strength can also be evaluated. Therefore, improving the distribution of the CFs in a composite can help to enforce the bond of CFs with the resin. Furthermore, as reported in our earlier work [3,28], sizing CFs with CNTs/CNFs can also modify the interaction between the CFs and resin. For example, sizing with 1 wt% of evenly distributed CNTs on CFs can increase the IFSS between CFs and resin to 89.0 ± 4.7 MPa, because the carbon nanofillers can enhance the spreading of the resin, and thus ensuring the processability of the fibres during infusion and leading to increased adhesion of the CFs with the polymeric matrix. This is also in agreement with Medina’s finding that 0.3 wt% of CNT incorporation can increase the IFSS between glass fibre and the resin by 19% [14].

### 3.3. Response of Shape Memory Polymer

Shape memory polymers can change to a predefined shape from a temporary shape under an appropriate stimulus such as temperature or stress. Due to their weak mechanical strength, CFRP was used to strengthen the SMP structure for an aerospace application.

As shown in Figure 13, the SMPC structures consisted of two CFs-reinforced prepreg layers with the interposition of one SMP foam interlayer. Nanoindentation was used to probe the response of the SMP interlayer in a smart SMPC. As shown in Figure 14a, with an increase of the peak load, the displacement increased, and the impressions on the shape memory layer became larger (Figure 15a). After heating to 110 °C, the impressions indented at room temperature all disappeared (Figure 15b) indicating the shape memory effect was invoked at this temperature. The indentation test was also carried out at 110 °C, and the displacement increased fast with the load initially; however, it became much slower with the increment of the load after 12 mN, indicating the SMP was recovering and counteracted on the load at this temperature (Figure 14b). As shown in the curves, although the final displacement was deeper than those indented at room temperature, the impressions disappeared due to free strain recovering in the retaining high-temperature period after unloading before cooling down [19]. In the meantime, the displacement was smaller than the indentation pressed at 100 °C, which suggested no shape memory response was triggered at 100 °C. After both the indenter and the sample were cooled to room temperature, different impressions were indented on the SM layer again (Figure 15c). Following that, the sample stage was heated to 100 °C and cooled down again; the size of the impressions reduced but they were not fully recovered as seen in Figure 15d. A similar temperature-induced shape recovery of the indentations was studied using atomic force microscopy [18]. This proved that the shape memory response temperature was between 100 and 110 °C for this shape memory foam. By adopting this strategy, the response temperature and behaviour of different shape memory materials can be accurately defined in a narrow range.

## 4. Conclusions

In this paper, different techniques developed on the Micro Materials nanoindentation instrument were used to characterize three smart composites for next-generation applications.

CNTs were integrated into the PPV2 matrix to increase the sensing and responding capability of the composite to develop a smart conveyor/impeller fan for air treatment. The non-uniformity due to structural anomalies or changes in properties at joints and boundaries was examined by a mapping nanoindentation test. It was found that the nanohardness and reduced elastic modulus of the composites produced via injection moulding had small variation with the change of the thickness and location, especially at the edge of the thicker plate.

Nanoindentation can be used to assess the mechanical properties of the CFs and the resin, as well as the interface between them. A push-out method developed on nanoindentation can be used to evaluate the interfacial properties between the CFs and resin, and therefore to examine the impact of the modification on CFs on their interfacial properties. Adding a small amount of carbon nanofillers such as CNTs into the sizing agents for CFs increased the bond strength between CFs and the polymer matrix significantly.

Nanoindentation can probe the in situ response of shape memory foam in the SMPC developed for aerospace application by indentations at room temperature and elevated temperatures. With the increment of the temperature, a large deformation was found at the same load; however, once a temperature was reached that invoked the shape memory response, a more rigid load–displacement curve was seen, which leads to a shallower impression. Furthermore, the triggering temperature range of the shape memory response can be determined by examining the change of indentations before and after a heating and cooling cycle.

## Figures and Tables

**Figure 1 polymers-14-02945-f001:**
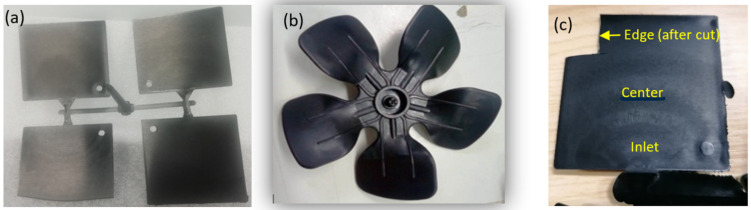
CNTs-PPV2 plates produced by injection moulding: (**a**) plates with different thicknesses; (**b**) an axial impeller fan; and (**c**) test part in one of the plates.

**Figure 2 polymers-14-02945-f002:**
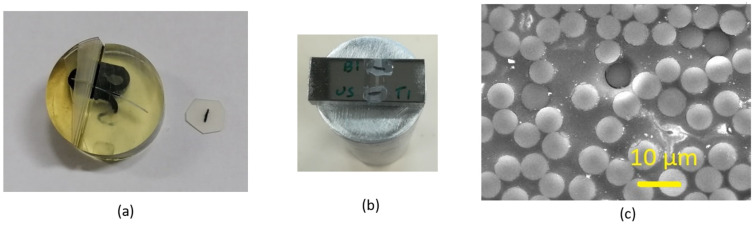
(**a**) The disc cut from the mounted composite, (**b**) specimen and holder for POT, (**c**) SEM image of the CFs in the CFRP (after POT test).

**Figure 3 polymers-14-02945-f003:**
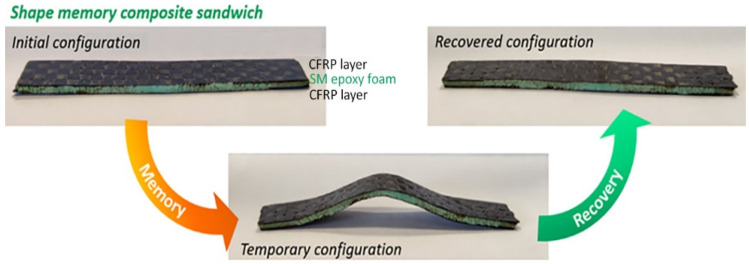
Shape memory composite sandwich structure.

**Figure 4 polymers-14-02945-f004:**
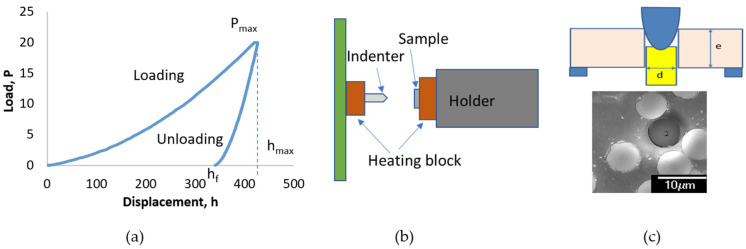
(**a**) Scheme of the indenter load versus its displacement during loading and unloading; (**b**) arrangement of the indenter and the sample on the heating stage in the Micro Materials nano-indentation; (**c**) schematic of the cone-shaped indenter in a push-out test (POT) setting and a CF after POT.

**Figure 5 polymers-14-02945-f005:**
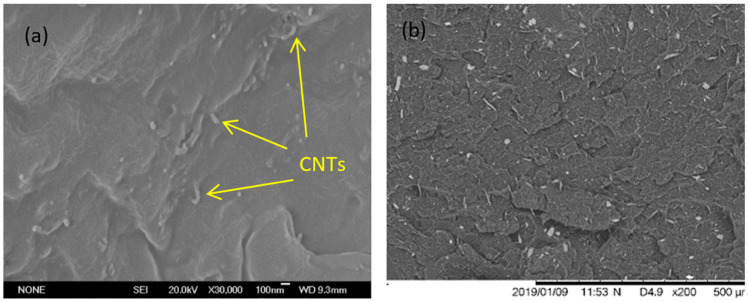
SEM observation of the CNTs-PPV2 nanocomposite (**a**) details of CNTs and (**b**) surface appearance.

**Figure 6 polymers-14-02945-f006:**
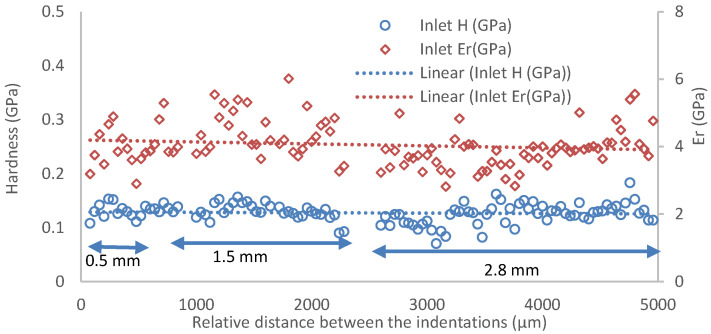
The nanohardness (H) and calculated reduced elastic modulus (Er) at the inlet section of the plates, where 0.5, 1.5, and 2.8 mm are the plates’ thicknesses.

**Figure 7 polymers-14-02945-f007:**
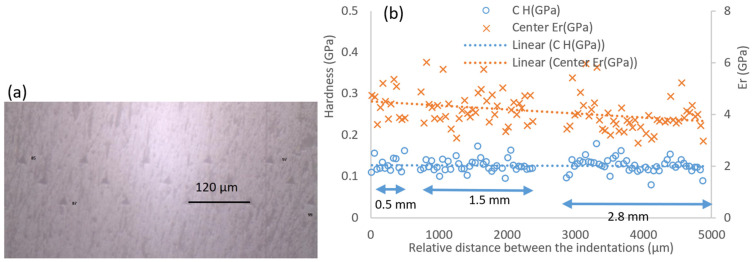
(**a**) Typical indentations and (**b**) nanohardness and calculated reduced elastic modulus at the centre area of the plates, where 0.5, 1.5, and 2.8 mm are the plates’ thicknesses.

**Figure 8 polymers-14-02945-f008:**
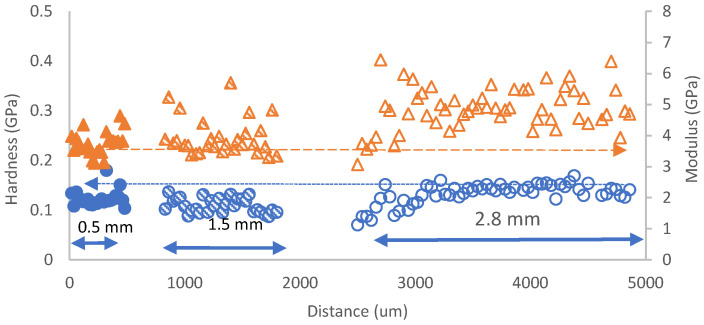
Nanohardness and calculated reduced elastic modulus at the edge area of the plates, where 0.5, 1.5, and 2.8 mm are the plates’ thicknesses (Hardness as circles and Er as triangles).

**Figure 9 polymers-14-02945-f009:**
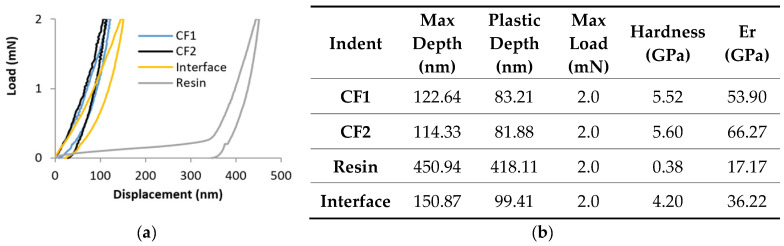
Nanoindentation test on CFRP: (**a**) load versus displacement curves for the interface carbon fibre/resin region, the resin and two carbon fibres (CF1 and CF2), and (**b**) the detailed values of each indentation.

**Figure 10 polymers-14-02945-f010:**
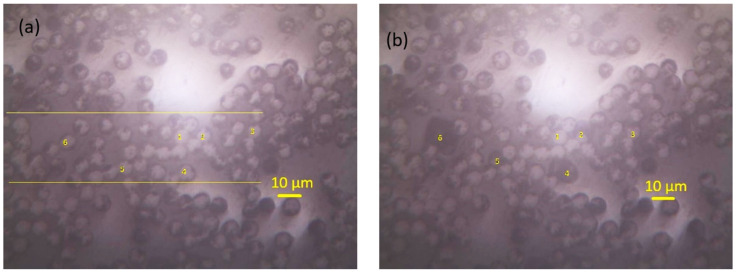
Selected region and the identified CFs in a CFs-reinforced composite (**a**) before indentation, (**b**) after indentation.

**Figure 11 polymers-14-02945-f011:**
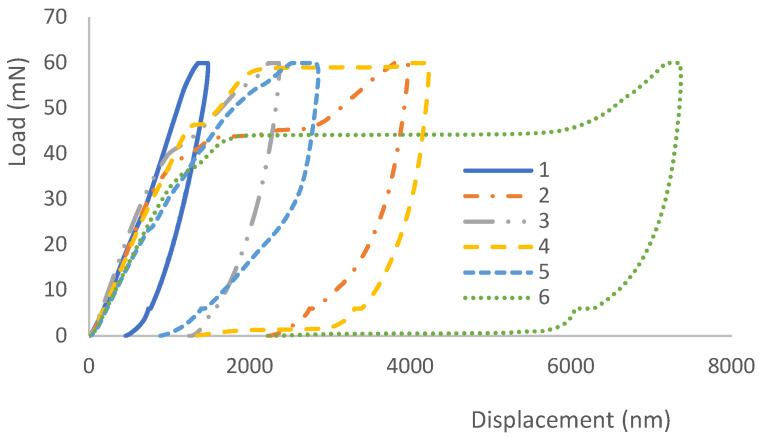
The loading–unloading vs. displacement curves for different CFs in the CFs-reinforced composite displayed in Figure 10.

**Figure 12 polymers-14-02945-f012:**
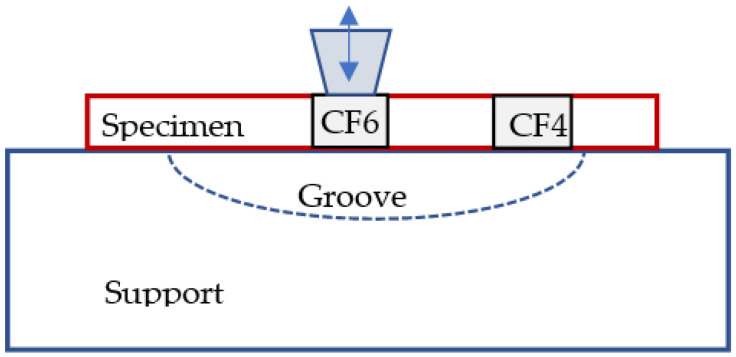
Impact of the location of CFs on top of the groove to the POT.

**Figure 13 polymers-14-02945-f013:**
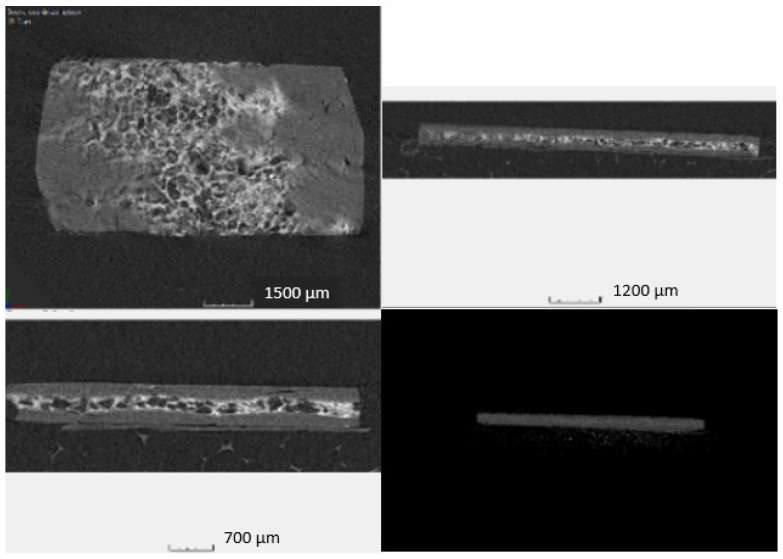
Micro-CT scan views from 3 directions on the shape memory composite.

**Figure 14 polymers-14-02945-f014:**
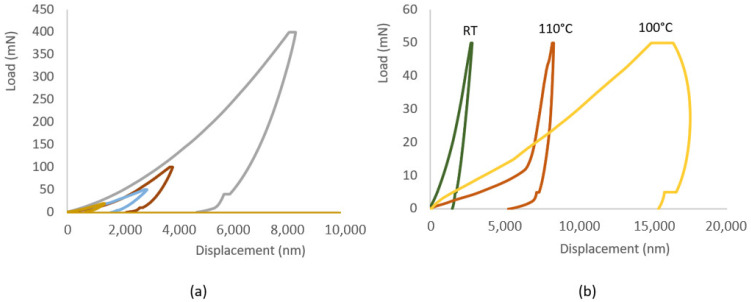
(**a**) Load versus displacement at room temperature (RT) with different peak loads (20, 50, 100, 400 mN), and (**b**) load against displacement curves at different temperatures: RT, 100 °C and 110 °C.

**Figure 15 polymers-14-02945-f015:**
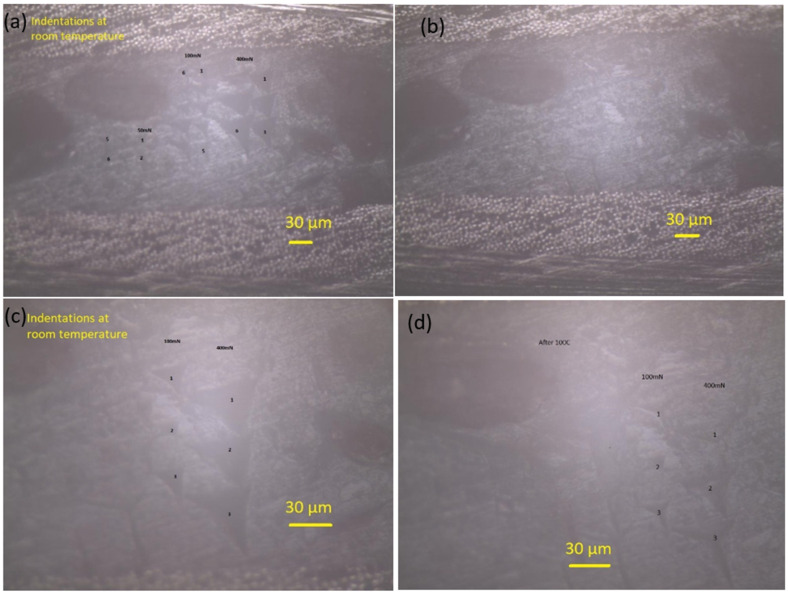
Indentation changes on SMPC: 50, 100, and 400 mN impressions, respectively depicted as 1, 2 and 3, (**a**) at room temperature; (**b**) after heating to 110 °C and cooling down; and 100 and 400 mN impressions (**c**) at room temperature; (**d**) after heating to 100 °C and cooling down.

**Table 1 polymers-14-02945-t001:** Typical nanoindentation application in evaluating polymer composites.

Composite Materials	Properties Characterization	Methods or Indentation	References
Homogeneous or heterogeneous composite, i.e., polymer nanotube composite	Evaluate nanomechanical properties including elastic modulus and nanohardness	Grid indentation or modulus mapping	[4,8,9]
Chemical modification of graphene-reinforced composite	Impact of the modified graphene on the resistance of the laminate under shear stress conditions	Push-in tests	[12]
An autoclave-cured CF-reinforced polymer plates	Effect of moisture absorption on the reduction of its stiffness due to the ageing in water	A flat-top cylinder indenter	[13].
Glass fibre epoxy composites	Influence of the CNT content on the glass fibre/matrix interfacial shear strength	Push-in and push-out test by a diamond flat conical tip	[14]
CNFs sized CFs-modified composite	Effect of sizing with CNFs on the interfacial properties of CFs and resin	Cone-shaped indenter push-out	[3]
CFs-reinforced SiC composite	Properties of the transversal and longitudinal cross-sections of individual CFs	Berkovich indenter indentation	[15]
CNTs-reinforced polymer composites	Nanomechanical (hardness/elastic modulus) and nanotribological (coefficient of friction) properties	Berkovich indenter	[16]
Unidirectional CFs-reinforced composite	Creep behaviour of the carbon fibre in the composite using POT technique	Berkovich and cone-shaped diamond indenters	[17]
Shape memory polymer	Deformation of the polymer networks at ambient and elevated temperatures, and modelling the shape memory response	Indentation load–depth response using Berkovich indenter	[18,19]

**Table 2 polymers-14-02945-t002:** The properties of the CNTs-reinforced PPV-2 composite.

Property	Unit	Value	STD Reference
Density	g/cm^3^	0.93	ASTM D 1505
Melt Index @ 230 °C 2.16/kg	g/10′	29 ± 2	ASTM D 1238
Flexural modulus	MPa	1800	ASTM D 790
HDT @ 0.455 MPa	°C	111	ASTM D 648
VICAT @ 10 N	°C	152	ASTM D 1525
Flame Resistance	Class	V-2 starting from 1.4 mm	UL-94

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
