# Peer review of "Nanoindentation of Multifunctional Smart Composites"

_polymers, 2022, doi:10.3390/polym14142945_

Round 1

Reviewer 1 Report

The manuscript (polymers-1804127) characterized composites using nanoindentation test through different ways, including mapping, pulling, and temperature variations. Overall, this work provides some practical usages of nanoindentation test, but lacks of deep investigation. I am wondering that whether authors could further explore the Nano-testing method.

1.       For CNT-PPV2 composites, Figures 7B and 8 show that the measured values also depend on plate thickness, what is the reason? Materials or tests? How about CNT dispersion in the matrix?

2.       For interfacial stress of CFRP, the data suggests that the adjacent carbon fibers do affect the pulling stress, and these data could provide extra information other than macroscopic measurements. Could author further explore the effects of adjacent carbon fibers on IFSS?

Author Response

Q: The manuscript (polymers-1804127) characterized composites using nanoindentation test through different ways, including mapping, pulling, and temperature variations. Overall, this work provides some practical usages of nanoindentation test, but lacks of deep investigation. I am wondering that whether authors could further explore the Nano-testing method.

R: Thanks for the suggestions. In this paper, we are focusing on introducing the versatility of nano-instrument on the characterisation of the different smart composites therefore it is difficult to give too much detail. However, we have done some extensive research on relevant topic such as:

"Evaluation of the creep behaviour of the carbon fibre in an unidirectional pultruded reinforced composite using nano-indentation technique, https://doi.org/10.1016/j.polymertesting.2019.106091"

"Viscoelastic response of carbon fibre reinforced polymer during push-out tests https://doi.org/10.1016/j.compositesa.2018.06.003"

  1. For CNT-PPV2 composites, Figures 7B and 8 show that the measured values also depend on plate thickness, what is the reason? Materials or tests? How about CNT dispersion in the matrix?

R1: The measured values show some difference with the plate thickness and location which might be linked to the pressure in the inject moulding process. By adjusting and optimising the inject moulding process parameter, the difference can be minimised which is the aim of our future work. We have rewritten the final words: “The nano-mechanical test suggests the impellers produced using the compound injected with a standard machine can get a uniform mechanical property by optimising the process parameters”.

The dispersion of CNTs in the matrix is important, and the agglomeration can lead to a local increase of hardness which we tried to avoid.

  1. For interfacial stress of CFRP, the data suggests that the adjacent carbon fibers do affect the pulling stress, and these data could provide extra information other than macroscopic measurements. Could author further explore the effects of adjacent carbon fibers on IFSS?

R2: Thanks for your suggestion, we have done some extensive research on this topic, we will carry on doing this in the future.

Zhang, Z.; Li, X.; Jestin, S.; Termine, S.; Trompeta, A.-F.; Araújo, A.; Santos, R.M.; Charitidis, C.; Dong, H. The Impact of Carbon Nanofibres on the Interfacial Properties of CFRPs Produced with Sized Carbon Fibres. Polymers 2021, 13, 3457. https://doi.org/10.3390/polym13203457

Liang, Y., Li, X., Giorcelli, M., Tagliaferro, A., Charitidis, C., & Dong, H. (2022). Enhancement and evaluation of interfacial adhesion between active screen plasma surface-functionalised carbon fibres and the epoxy substrate. Polymers, 14(4), [824]. https://doi.org/10.3390/polym14040824

Reviewer 2 Report

The paper shows results about nano-indentation multifunctional smart composites. Authors studied some multifunctional composites for next-generation applications using nano-indentation investigation techniques. They  are used to determine peculiarities and specific properties for the different composites. Mentioned composite it suitable to be used in axial impeller fans and the results can be used to improve the process of the composite produced by injection moulding. The interfacial properties of the carbon fiber (CF) and the resin were evaluated by a push-out method utilizing the smaller indentation tip to target the individual CF and apply load to measure its displacement under loads. It can be used in order to evaluate the effectiveness of the surface modification on the CFs, like sizing. Authors used nano-indentation at different temperatures for the probing of the in-situ response of smart shape memory polymer composite (SMPC) usable in grabbing devices for aerospace applications.

Dear authors. Thank you very much for your value paper about nano-indentation multifunctional smart composites. The results can find real applications. I have some comments and suggestions, what should be considered in presented to review paper.

Comments and suggestions:

1. Chapter Introduction well describes fundamental information about nano composites, where role of carbon nanotubes and other nano fibers is clear presented. Then, authors explain nano identification procedure, which is based on electromagnetic and capacitive phenomena.

2. Presented in Table 1 composites and their properties, based on rich references are complete.

3. Title and subtitle words should start using capital letters. Please correct.

4. Chapter 2.1. – I could not find any information about the sizes, in nano meter, of nano particles used in studied materials. Please complete the information.

5. Fig. 8 – please differentiate which points are related to which axis. It is difficult to recognize it.

Author Response

Dear authors. Thank you very much for your value paper about nano-indentation multifunctional smart composites. The results can find real applications. I have some comments and suggestions, what should be considered in presented to review paper.

Comments and suggestions:

  1. Chapter Introduction well describes fundamental information about nano composites, where role of carbon nanotubes and other nano fibers is clear presented. Then, authors explain nano identification procedure, which is based on electromagnetic and capacitive phenomena.
  2. Presented in Table 1 composites and their properties, based on rich references are complete.
  3. Title and subtitle words should start using capital letters. Please correct. Thanks to the reviewer for the suggestion, we corrected the formatting.
  1. Chapter 2.1. – I could not find any information about the sizes, in nano meter, of nano particles used in studied materials. Please complete the information. The CNTs have mean outer diameter of 10-15 nm with length between 0.1-10 µm, and this information has been added now as follow: “the mean outer diameter is 10-15 nm with length between 0.1-10 µm)”.
  2. Fig. 8 – please differentiate which points are related to which axis. It is difficult to recognize it. We adjusted the colour of the data and used two arrows to indicate the data and make it clear.

Round 2

Reviewer 1 Report

Evaluation of the creep behaviour of the carbon fibre in an unidirectional pultruded reinforced composite using nano-indentation technique, https://doi.org/10.1016/j.polymertesting.2019.106091

Viscoelastic response of carbon fibre reinforced polymer during push-out tests https://doi.org/10.1016/j.compositesa.2018.06.003

Zhang, Z.; Li, X.; Jestin, S.; Termine, S.; Trompeta, A.-F.; Araújo, A.; Santos, R.M.; Charitidis, C.; Dong, H. The Impact of Carbon Nanofibres on the Interfacial Properties of CFRPs Produced with Sized Carbon Fibres. Polymers 2021, 13, 3457. https://doi.org/10.3390/polym13203457

Liang, Y., Li, X., Giorcelli, M., Tagliaferro, A., Charitidis, C., & Dong, H. (2022). Enhancement and evaluation of interfacial adhesion between active screen plasma surface-functionalised carbon fibres and the epoxy substrate. Polymers, 14(4), [824]. https://doi.org/10.3390/polym14040824

These papers have done many works on measuring interfacial stress of CFRP. What is the advance of current work as compared with above works? If authors want to introduce the versatility of nano-instrument on the characterisation of the different smart composites, please write a review article instead of a research paper. As a research paper, authors should focus on new or potential applications instead of common usages.

Also, the authors state that “The measured values show some difference with the plate thickness and location which might be linked to the pressure in the inject moulding process” The questions are: Is the difference significant or not? What changes of the structure cause the difference? And what processing parameters cause the structural changes?

Author Response

Evaluation of the creep behaviour of the carbon fibre in an unidirectional pultruded reinforced composite using nano-indentation technique, https://doi.org/10.1016/j.polymertesting.2019.106091

Viscoelastic response of carbon fibre reinforced polymer during push-out tests https://doi.org/10.1016/j.compositesa.2018.06.003

Zhang, Z.; Li, X.; Jestin, S.; Termine, S.; Trompeta, A.-F.; Araújo, A.; Santos, R.M.; Charitidis, C.; Dong, H. The Impact of Carbon Nanofibres on the Interfacial Properties of CFRPs Produced with Sized Carbon Fibres. Polymers 2021, 13, 3457. https://doi.org/10.3390/polym13203457

Liang, Y., Li, X., Giorcelli, M., Tagliaferro, A., Charitidis, C., & Dong, H. (2022). Enhancement and evaluation of interfacial adhesion between active screen plasma surface-functionalised carbon fibres and the epoxy substrate. Polymers, 14(4), [824]. https://doi.org/10.3390/polym14040824

Q1) These papers have done many works on measuring interfacial stress of CFRP. What is the advance of current work as compared with above works? If authors want to introduce the versatility of nano-instrument on the characterisation of the different smart composites, please write a review article instead of a research paper. As a research paper, authors should focus on new or potential applications instead of common usages.

R1) Authors are sorry that the novelty of the study has not been fully described, yet. The main achievements are related to the application of the proposed technique (already published in previous works) on shape memory polymer composites. Testing of CNT filled thermoplastics has been carried out for comparison, for a better understanding of the adopted experimental procedure. For this aim, a new reference has been added, and now, it is reported in the study: “For the first time, in this work, SMPCs have been tested under nano-indentation whereas, in a previous study, the sole SMP matrix was tested under micro-indentation [26]. It was found that SMP samples exhibited very high recovery capabilities of the indented marks. Nevertheless, it was not clear if this behavior was improved by the foam nature of the SMP samples. Thanks to nano-indentation, the same recovery mechanism has been observed at a lower scale, and novel experimental results are discussed. For comparison, tests have been also performed on typical CNT filled thermoplastics, for which the same nano-indentation techniques were already used in previous studies. This comparison allows providing a wider scenario for the application of the proposed experimental procedure”.

Q2) Also, the authors state that “The measured values show some difference with the plate thickness and location which might be linked to the pressure in the inject moulding process” The questions are: Is the difference significant or not? What changes of the structure cause the difference? And what processing parameters cause the structural changes?

R2) Authors are sorry for the misunderstanding about this effect of the thickness. The effect of the thickness is significant and expected. Now it is discussed: “Nevertheless, the effect of the plate thickness is present, and is related to the polymer matrix and filler orientation during injection. In fact, because of the fountain flow, polymer structures in the skin, close to the mould walls, are elongated by extensional flow whereas orientation in the core is by shear. As a result, CNTs are orientated along the plate face in the skins, and along the thickness in the core. By reducing the plate thickness, the contribution of the extensional flow becomes more and more important, and this effect is recognized by the nano-indentations.”

Round 3

Reviewer 1 Report

The manuscript has been properly revised.